# Gaucher Disease in Internal Medicine and Dentistry

Michele Basilicata [1,2,†], Giulia Marrone [3,†], Manuela Di Lauro [3,*], Eleonora Sargentini [3], Vincenza Paolino [1,3], Redan Hassan [4], Giuseppe D'Amato [5], Patrizio Bollero [1,3] and Annalisa Noce [3,6,*]

1 UOSD Special Care Dentistry, Policlinico Tor Vergata, 00133 Rome, Italy
2 Department of Experimental Medicine and Surgery, University of Rome Tor Vergata, 00133 Rome, Italy
3 Department of Systems Medicine, University of Rome Tor Vergata, 00133 Rome, Italy
4 General Surgery and Organ Transplantation Unit, Department of General and Specialistic Surgery "Paride Stefanini", AOU Policlinico Umberto I, University of Rome Sapienza, 00185 Rome, Italy
5 Unicamillus International University of Health and Medical Sciences, 00131 Rome, Italy
6 UOSD Nephrology and Dialysis, Policlinico Tor Vergata, 00133 Rome, Italy
* Correspondence: manuela.dilauro@alumni.uniroma2.eu (M.D.L.); annalisa.noce@uniroma2.it (A.N.); Tel.: +39-06-20902191 (M.D.L.); +39-06-20902194 (A.N.)
† These authors are equally contributed to this work.

**Abstract:** Gaucher disease (GD) is a lysosomal storage pathological condition, characterized by a genetic autosomal recessive transmission. The GD cause is the mutation of GBA1 gene, located on the chromosome 1 (1q21), that induces the deficiency of the lysosomal enzyme glucocerebrosidase with consequent abnormal storage of its substrate (glucosylceramide), in macrophages. The GD incidence in the general population varies from 1:40,000 to 1:60,000 live births, but it is higher in the Ashkenazi Jewish ethnicity (1:800 live births). In the literature, five different types of GD are described: type 1, the most common clinical variant in Europe and USA (90%), affects the viscera; type 2, characterized by visceral damage and severe neurological disorders; type 3, in which the neurological manifestations are variable; cardiovascular type; and, finally, perinatal lethal type. The most affected tissues and organs are the hematopoietic system, liver, bone tissue, nervous system, lungs, cardiovascular system and kidneys. Another aspect of GD is represented by oral and dental manifestations. These can be asymptomatic or cause the spontaneous bleeding, the post oral surgery infections and the bone involvement of both arches through the Gaucher cells infiltration into the maxilla and mandibular regions. The pharmacological treatment of choice is the enzyme replacement therapy, but the new pharmacological frontiers are represented by oral substrate reduction therapy, chaperone therapy, allogeneic hematopoietic stem cell transplantation and gene therapy.

**Keywords:** Gaucher disease; bone; jaw; kidney; eruptive delay; radiographic signs; oral health; Gaucher cells; neurological disorders; splenomegaly





## 1. Introduction

Gaucher disease (GD) is a genetic condition determined by the autosomal recessive mutation of GBA1 gene, located on the chromosome 1 (1q21). This mutation leads to the deficiency of the lysosomal enzyme glucocerebrosidase (GCase) and to the consequent abnormal storage of its substrate, glucosylceramide (GlcCer), in macrophages. GCase metabolizes the glicosphyngolipid glucosylceramide into ceramide and glucose; its lack of activity leads to the accumulation of its substrates and to the transformation of macrophages into Gaucher cells (GCs). GCs have a larger size, eccentric nuclei, condensed chromatin and cytoplasm with the aspect of wrinkled tissue paper. GCs not only result from the transformation of macrophages, but also seem to represent a M2 subpopulation. Less frequently, GD is determined by the mutation of the GCase activator, the saposin C; this variant implies the same pathways of the GCase mutation, which is more frequent.

The GD incidence in the general population varies from 1:40,000 to 1:60,000 live births, but it is higher in the Ashkenazi Jewish ethnicity, in which it rises up to 1:800 live births. In

the literature, five different types are enumerated according to the clinical features: type 1, which represents the most common clinical variant in Europe and USA (90%), and it affects the viscera; type 2, in which the visceral damage is associated with severe neurological disorders; type 3, in which the neurological manifestations are variable and may not be present; cardiovascular type; and, finally, perinatal lethal type.

More than 300 mutations in the GBA1 gene have been described in the literature [1,2]. The four most common mutations identified are N370S, IVS2, 84GG and L444P [3]. The main mechanism in the pathogenesis of GD is the infiltration of many tissues and organs by GCs (Figure 1).

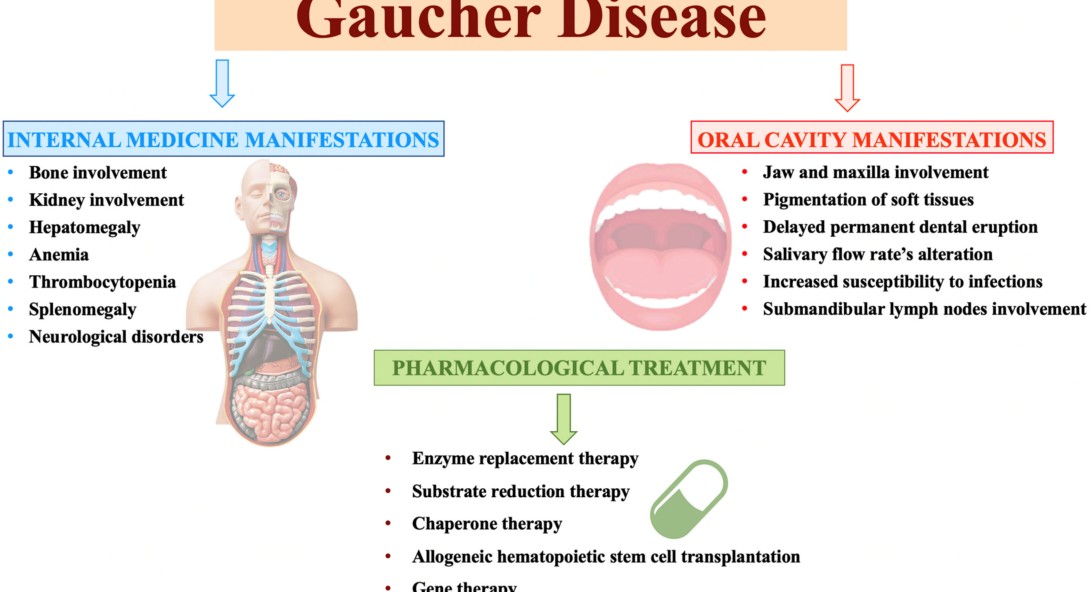

**Figure 1.** Gaucher disease manifestations in internal medicine and dentistry, including pharmacological treatment.

GCs mainly infiltrate bone marrow, the spleen and the liver. The lack of enzyme activity may have effects on numerous cells, among these, hematopoietic progenitor cells, erythrocytes, mesenchymal cells, hepatocytes and nervous cells [1].

GD is a storage disorder and is characterized by systemic manifestations, involving different organs and apparatus, among which there is the stomatognathic system. In particular, lesions of the jaw, incidentally detected by jaws or intraoral dental X-rays, can represent the first sign of the GD and often help in its diagnosis. Therefore, the aim of this review is to discuss the clinical manifestations of GD in internal medicine, particularly in the nephrology field, and in dentistry, thus describing the possible pharmacological treatments.

## 2. Materials and Methods

We have analyzed several articles selected from scientific databases such as PubMed, Web of Science, Scopus and Embase. We selected the papers containing keywords such as "Gaucher disease" in combination with "internal medicine" or "dentistry" and "bone involvement" and/or "chronic kidney disease" and/or "radiographic signs" and/or "kidney disease" and/or "oral health" and/or "bisphosphonates" and/or "special care dentistry". In our review, we inserted observational, case control and case report studies. The references list and the related records were manually reviewed by the authors. The search was limited to English language papers published until December 2022.

### 3. GD in Internal Medicine

The GD clinical signs and symptoms are several and involve many organs and apparatus. The systemic manifestations should be divided into hematological, visceral, neurological, skeletal and metabolic. In this section, we describe the main findings of GD (Table 1).

- *Hematopoietic system*

Among the apparatus and organs affected by GD, the hematopoietic system is widely involved. In this regard, the most common early signs are represented by the anemia and the thrombocytopenia. Another important sign is the splenomegaly, which comes from the suppression of hematopoiesis due to the deposits of GCs in bone marrow, which leads also to pancytopenia. In addition, monoclonal and polyclonal gammopathies are described. These manifestations expose the patient to a higher risk of developing myeloma. Therefore, the immune dysregulation may create a favorable environment for the solid cancer and hematological neoplasms onset [3]. Furthermore, in GD patients are described acquired and inherited alterations in the coagulation system, especially coagulation factors deficiencies. Dysfunctional platelet aggregation and low-grade disseminated intravascular coagulation (DIC) have been reported [4].

- *Liver*

Another organ commonly affected in GD is the liver, and hepatomegaly is one of the manifestations. The increase in the hepatic and biliary markers is often observed. The long-term hepatic complications of GD are fibrosis, cirrhosis and portal hypertension, which represent high-risk factors for hepatocarcinoma (HCC). The main processes at the base of the onset of HCC are various abnormalities, such as the alterations in iron metabolism, the insulin resistance, the immune dysregulation, the endoplasmic reticulum stress, the splenectomy, etc. [5].

- *Nervous system*

Individuals with type 2 or 3 GD are affected by severe neurological disorders. The pyramidal system involvement is often encountered. It appears with classic signs, such as opisthotonus, head retroflexion, spasticity and trismus. Moreover, bulbar signs, such as stridor, squint and swallowing difficulty, are also present. Other neurological manifestations, which can occur, are: the oculomotor apraxia, the saccadic initiation failure, the opticokinetic nystagmus, the generalized tonic-clonic seizures and the progressive myoclonic epilepsy [4]. Patients with heterozygous or homozygous GBA1 mutation are more at risk for developing an early Parkinson's disease. The GCase deficiency leads to the lack of $\alpha$-synuclein metabolism. Consequently, the accumulation of $\alpha$-synuclein results in neurotoxicity, especially in the substantia nigra. In addition, GlcCer, namely, a GCase's substrate, stabilizes the oligomers of $\alpha$-synuclein, which in turn create aggregates, called Lewy bodies, in nervous cells [1].

- *Cardiovascular system*

Among the different GD types previously described, there is the cardiovascular form, in which a specific genotype (p.Asp448His allele) corresponds to a rare phenotype with prevalent cardiovascular clinical manifestations, mainly calcification of the mitral and aortic valves [4].

- *Pulmonary apparatus*

Even if it is considered very rare, the pulmonary implication in GD is reported. The main patterns of lung damage identified are: the thickening of the interlobular and intralobular septa, the opacities in the alveoli and intercellular space and the blockage of capillaries. Furthermore, fibrosis of the pulmonary tissues and engagement of the hilar and mediastinal lymph nodes have been reported [6].

- *Metabolic pathways*

GD is also associated with metabolic issues. It has been stated that GD patients register an excessive energy expenditure, probably related to the macrophage activation and to the increased release of pro-inflammatory cytokines [5,7].

It was also assumed that GD patients were more predisposed to register an increase in body weight and in fat mass, due to the enzymatic replacement treatment (ERT) [7]. On the contrary, a study conducted by Langeveld et al. suggested that the body weight gain of GD subjects was not related to the ERT, but to the physiological aging [8]. In fact, the trend in body weight gain seems to be more related to the enhanced caloric intake and to the sedentariness [5].

Moreover, GD is related to insulin resistance. GM3 is one of the sphingolipids that seems to be associated with the interference in insulin signaling by (i) relocating insulin receptor in lipid raft domains, by (ii) compromising insulin receptor interaction and by (iii) lowering one of the pathways of insulin signaling (PI(3)K/Akt) [9]. GD patients register an increase in the glucose production, a reduction in the glucose clearance and, thus, higher levels of insulin [7,10]. It has been hypothesized that insulin resistance is not related to overweight caused by ERT, because it was also observed in non-overweight GD subjects on ERT [5,11].

Therefore, alterations in lipid metabolism and trafficking, affecting gangliosides (GM3), phosphatidylcoline and sphingomyelin, were observed [12]. In addition, a decrease in blood levels in total, low-density lipoprotein (LDL) and high-density lipoprotein (HDL) cholesterol, in apolipoprotein B (ApoB) and apolipoprotein AI (ApoAI), and an increase in plasma concentrations of triglycerides and apolipoprotein E (ApoE) were reported [13,14].

**Table 1.** GD manifestations and internal medicine.

| Type of the Study | Authors | Year | Number of Patients | Description of the Study | Findings |
|---|---|---|---|---|---|
| Observational | Hollak et al. [7] | 1997 | 12 GD type 1 adult patients | Assessment of: REE Liver and spleen volume Hemoglobin Platelet count before and after 6 months of alglucerase therapy | ↑ Glucose production in GD patients ↓ REE ↑ Body Weight ↑ Hemoglobin and platelet count ↓ Liver and spleen volumes |
| Observational | Langeveld et al. [8] | 2008 | 35 type I GD patients on ERT 7 untreated type I GD patients | Evaluation of: Insulin resistance Type II diabetes prevalence | 6% of prevalence of IR in ERT patients 8.2% of prevalence of DMT2 in ERT patients 0% of prevalence of IR in untreated patients 0% of prevalence of DMT2 in untreated patients |
| Cross-sectional | Ucar et al. [11] | 2009 | 14 non-overweight GD patients treated with ERT | Measurement of insulin resistance by HOMA-IR OGTT | 6.6% of prevalence of IR ↑ HOMA-IR compared to healthy controls ↑ post-loading glucose compared to healthy controls ↑ fasting and postloading insulin compared to healthy controls |

**Table 1.** *Cont.*

| Type of the Study | Authors | Year | Number of Patients | Description of the Study | Findings |
|---|---|---|---|---|---|
| Observational | Meikle et al. [12] | 2007 | 41 type I GD patients (pre- and post-therapy) 30 controls | Evaluation of: The role of plasma sphingolipids and phospholipids as biochemical markers of the efficacy of GD pharmacological treatment | ↑ levels of: Glucosylceramide Molecular species of phosphatidylglycerol G(M3) ganglioside ↓ levels of: Ceramide species, Dihexosylceramide Sphingomyelin |
| Observational | Ginsberg et al. [13] | 1984 | 29 type 1 GD patients | Assessment of: Plasma lipid concentrations Serum apoprotein concentrations | ↓ plasma total cholesterol ↓ LDL cholesterol ↓ HDL cholesterol ↓ serum levels of: apoprotein-B and apoprotein-AI ↑ serum levels of: apoprotein-E |
| Observational | Pocovi et al. [14] | 1998 | 258 subjects: 57 GD patients 137 non-affected carriers of mutations 64 non-carriers | Assessment of: The link between mutations of the glucocerebrosidase locus and hypo-alpha-lipoproteinemia | Heterozygosity for the glucocerebrosidase mutations: ↓ HDL-C Affected subjects ↓ LDL-C ↓ HDL-C compared to non-affected carriers and non-carriers Different HDL-C plasma concentration between non-affected and non-carriers |

Abbreviations: DMT2, diabetes mellitus type 2; ERT, enzyme replacement therapy; GD, Gaucher Disease; HDL, high density lipoprotein; HOMA-IR, Homeostasis Model Assessment; IR, insulin resistance; LDL, low density lipoprotein; OGTT, Oral Glucose Tolerance Test; REE, resting energy expenditure; ↓, Decreased; ↑, Increased.

- *Kidney involvement*

According to the literature, only few cases of renal involvement in GD were described [15,16] (Table 2). A small number of studies showed renal involvement when subjects were alive [15,17,18] instead; other kidney findings were detected during the autopsy [19,20].

In presence of kidney involvement, glomerular abnormalities were described. GCs infiltration causes tubular enlargements, capillary occlusion, mesangial cells and matrix proliferation. Moreover, the thickening of basement membranes and the sclerosis of many glomeruli were observed [15,21].

In two studies, the immunofluorescence was performed: the first by De Brito et al. [15] produced a negative result, while in the second one, Smith et al. [17] found IgM and IgA deposits within the capillaries.

In the electron microscope, Pennelli et al. discovered Gaucher's bodies inside mesangial and endothelial cells and the latter seem to take the form of GCs [19]. Extracellularly, these deposits were present in mesangial matrix.

Siegal et al. [20] described a case report of an Ashkenazi woman with GD who developed renal failure 24 years after the splenectomy. The biopsy of her kidneys post mortem showed the presence of GCs in the glomeruli, capillaries and in the interstitium of the cortex. Moreover, the glomeruli presented fibrosis, but it was also registered interstitial

fibrosis associated with tubular atrophy. The authors described also in mesangial and endothelial cells intracytoplasmatic accumulation of Gaucher bodies.

The pathogenetic role of splenectomy in renal involvement in GD was established by Matoth et al. [22], which hypothesized that the surgery procedure induced the sharp increase in plasma glucocerebrosidase levels with a consequent impact on the reticulo-endothelial cells.

On the other hand, in a study conducted by Becker-Cohen et al. [23] on 161 patients with GD, there was no evidence of renal involvement. The only significant result was the presence of hyperfiltration in GD patients compared to general population, matched for gender and age, both in adulthood and in pediatric subjects. In particular, the mean glomerular filtration rate (GFR) of many patients was more than two standard deviations greater than the one observed in the general population. These data could be partly explained by the increase in the angiotensin-converting enzyme activity, found in GD patients [24]. In fact, the enhanced activation of renin-angiotensin-aldosterone system causes a consequent hyperfiltration [25].

Kim et al. [26] described a case of a 45-year-old woman with a mass at the lower pole of the right kidney, which after the immunohistochemical analysis resulted to be a diffuse large B-cell lymphoma (DLBCL). The urinalysis findings were normal. The bone marrow biopsy showed the infiltration of histiocytes and/or macrophages, characterized by a cytoplasm with the appearance of a wrinkled tissue paper. This finding led the authors to identify them with GCs; moreover, the authors found the β-glucocerebrosidase enzyme deficiency. In this case report, the GbA gene analysis was performed, confirming a pathogenic mutation, G202R (c.721G>A; p.G241R), and a novel mutation R277C (c.946C>T; p.R316C), and thus allowing the authors to speculate the presence of GD.

The deposits of sphingolipids in the cells of GD patients seem to be related to immune dysregulation [27]. Furthermore, this alteration may lead GD patients to a higher risk of developing cancer of all kinds, but especially hematologic malignancies [26]. In fact, Shiran et al. [28] demonstrated that GD patients are 14.7 times more likely to develop hematologic malignancies and 3.6 times more likely to suffer from non-hematologic malignancies than the general population. On the other hand, Rosenbloom et al. [29], since the occurrence of hematologic malignancies among GD patients is 0.7%, hypothesized that this cancer is not so frequent. Therefore, the kidneys are rarely involved in DLBCL and even less in primary renal non-Hodgkin lymphoma [30]. In addition, renal involvement, at the time of the DLBCL diagnosis, has an incidence rate of only 2% [31].

Al-Bderat et al. [32] described the case of a 18-month child, born with a premature vaginal delivery due to gestational diabetes, who developed abnormal movements of arms and face and nephrotic syndrome (NS). The kidney biopsy was performed and showed the global sclerosis in eight glomeruli and segmental sclerosis in thirty glomeruli. When he was 4, he developed an end-stage renal disease and started a renal replacement therapy (hemodialysis). Due to the presence of a hepatosplenomegaly and a pancytopenia, the patient underwent bone marrow biopsy and the findings were suggestive of GD. The fibroblast test showed low glucocerebroside activity. Therefore, the authors described this as the first case of GD in a patient with a previous diagnosis of focal segmental glomerulosclerosis (FSGS). The association between GD and FSGS was also described in other few case reports in literature, above mentioned [17,20,33].

Another glomerulopathy correlated to GD is mesangiocapillary glomerulonephritis (MCGN). Halevi et al. [34] described the case of a 6-year-old child who presented NS and MCGN, documented through the biopsy. There was a suspicion of GD due to the finding of aseptic necrosis of the femoral heads on X-rays. The bone marrow test and, later, the test on white blood cells, confirmed the GD diagnosis.

The NS secondary to amyloidosis (AL) related to GD was described in a case report by Kaloterakis et al. [35]. A 47-year-old Greek man with a previous diagnosis of GD was admitted to the hospital with oedema in both legs, pancytopenia and NS. He also presented lymphadenopathy, liver enlargement, splenomegaly with areas of necrosis, deformity

at the distal part of the femurs and bone marrow infiltration by GCs. He underwent a splenectomy and a biopsy of the liver, the kidney and the spleen. During the surgery procedure, the clinicians revealed extensive amyloid depositions, which were classified as AL. Two years after the splenectomy, the patient developed anasarca, pleurisy and ascites. The bone marrow test revealed the infiltration of GCs and a diffuse plasmacytosis. The cardiac, renal, neurological, muscular and bone condition worsened, and the patient died 30 months after the splenectomy.

In the literature, three other cases of AL related to GD were described [36–38], but the patients died of heart failure soon after the AL diagnosis.

Currently, only two cases of NS, associated with GD and infiltration of kidneys by GCs without AL, are described [39].

**Table 2.** GD manifestations and chronic kidney disease.

| Type of the Study | Authors | Year | Number of Patients | Description of the Study | Findings |
|---|---|---|---|---|---|
| Case report | Siegal et al. [20] | 1981 | 1 GD patient | Ashkenazi woman with GD who developed renal failure after 24 years since the splenectomy | Presence of GCs in the glomeruli, capillaries and in the interstitium of the cortex Fibrosis of the glomeruli, tubular atrophy and interstitial fibrosis Mesangial and endothelial cells, intracytoplasmatic accumulation of Gaucher bodies |
| Observational | Becker-Cohen et al. [23] | 2005 | 161 GD patients | Renal involvement in GD | No evidence of renal involvement Presence of hyperfiltration in GD patients (male, female and pediatric groups) compared to general population |
| Case report | Kim et al. [26] | 2012 | 1 patient | 45-year-old woman with a mass at the lower pole of the right kidney (DLBCL) | Bone marrow biopsy: infiltration of histiocytes and/or macrophages (cytoplasm with the appearance of a wrinkled tissue paper) β-GCase enzyme deficiency GbA gene analysis: indentification of a pathogenic mutation, G202R, and a novel mutation R277C |
| Case-control | Shiran et al. [28] | 1993 | 51 GD patients 511 control subjects | Risk of developing cancer (especially hematologic neoplasms) | Developing of cancer: 10 out 48 GD patients 35 out of 511 control subjects GD patients had a risk 14.7 times greater of hematologic neoplasms |
| Registry | Rosenbloom et al. [29] | 2005 | 2742 young or middle-aged GD patients | Assessment of: The risk of cancer in comparison with the risk expected in the United States population of the same age and sex | 10 patients affected by myeloma (RR 5.9) Overall risk for cancer of all kinds: 0.79 |

**Table 2.** *Cont.*

| Type of the Study | Authors | Year | Number of Patients | Description of the Study | Findings |
|---|---|---|---|---|---|
| Case report | Al-Bderat et al. [32] | 2016 | 1 GD patient | Eighteen-month-old child, born with a premature vaginal delivery due to gestational diabetes, who developed abnormal movements of arms and face and NS | Biopsy findings: global sclerosis in eight glomeruli and segmental sclerosis in thirty glomeruli Bone marrow biopsy: infiltration by histiocytes with wrinkled tissue cytoplasm Fibroblast test results: low glucocerebroside activity |
| Case report | Halevi et al. [34] | 1993 | 1 GD patient | Six-year-old child with nephritic syndrome and MCGN (biopsy) | X-rays results: AVN of the femoral heads Bone marrow infiltration GCase activity deficiency in white blood cells |
| Case report | Kaloterakis et al. [35] | 1999 | 1 GD patient | Forty-seven-year-old Greek patient admitted to the hospital with oedema in both legs, pancytopenia and NS | Lymphadenopathy Liver enlargement Splenomegaly Deformity at the distal part of the femurs Bone marrow infiltration by GCs Extensive amyloid depositions) Two years after the splenectomy: Anasarca, pleurisy and ascites Infiltration of GCs and a diffuse plasmacytosis at the bone marrow test Cause of death: worsening of the cardiac, renal, neurological, muscular and bone condition |
| Case report | Hanash et.al [36] | 1978 | 1 GD patient | Forty-six-year-old woman GD patients with amyloidosis and 3100 mg/dl of monoclonal IgA | Cause of death: restrictive cardiac disease |
| Case report | Hrebicek et al. [37] | 1996 | 1 GD patient | Forty-seven-year-old GD patient with cardiopulmonary amyloidosis | Cardiac and pulmonary deposits of amyloid GCase deficiency Chitotriosidase deficiency Heterozygosity in the GCase gene (N370S and D409H mutations) Absence of GCs in lungs |
| Case report | Dikman et al. [38] | 1978 | 1 GD patient | Patient with NS and systemic amyloidosis | Bone marrow test: Plasmacytosis Light chains urine and glomeruli Low immunoglobulin serum levels |
| Case report | Morimura et al. [39] | 1994 | 1 GD patient | Thirty-three-year-old Japanese GD patient (cause of death: renal and pulmonary failure) | Autopsy: GCs infiltration in the liver, bone marrow, lymph nodes, kidneys and lungs |

Abbreviations: AVN, avascular necrosis; DLBCL, diffuse large B-cell lymphoma; GbA, glucosylceramidase; Gcase, glucocerebrosidase; GCs, Gaucher cells; GD, Gaucher Disease; MCGN, mesangiocapillary glomerulonephritis; NS, nephrotic syndrome.

- *Bone tissue involvement*

GCs seem to influence the hematopoiesis and the balance of the number and the activity of the osteoblasts and osteoclasts, inducing an impaired shaping of trabecular and cortical bone [40]. An element that may confuse the GD diagnosis is the presence of atypical variants of GCs, found in untreated patients [41]. Their effect on the bone tissue is a progressive and centrifugal displacement of normal adipocytes [42], initially into the axial skeleton and then into the extremities [43]. This infiltrative process sets early in the GD and it usually develops before the bone symptoms [44]. The pathogenesis is not entirely clear. However, in several studies, it was found that the glucocerebrosidase inhibition alters the processes of erythropoiesis, the myeloid proliferation and the differentiation/development of mesenchymal stem cells [45]. In addition, the expression profiles of cytokines and prostaglandins of bone marrow mesenchymal stromal cells are altered. These changes, due to the osteoclasts and plasma cells proliferation, not only reduce the bone mineral density (BMD), but also induce the onset of polyclonal and monoclonal gammopathies [46].

The manifestations of bone involvement in GD include bone pain or bone crisis, bone marrow infiltration, avascular osteonecrosis (AVN), infarction, pathological fractures, typical deformities, and an early osteopenia and osteoporosis. Bone pain and kyphosis are GD pathognomonic signs [47].

According to the literature, 80–95% of patients with GD1 have a bone involvement [48], and in the 25–32% of cases, it represents the only or the main manifestation of GD [49–53]. The detectable clinical signs of GD1 are Erlenmeyer's femur deformity resulting from metaphyseal enlargement of the bone, osteolytic lesions and bone crisis. They are characterized by severe bone pain, resulting from limited or localized bone infarcts. The latter sign may be a precursor of future osteonecrosis and fractures. [54,55].

Histologically, there is a very heterogeneous picture of infiltrated bone marrow by GCs, characterized by areas of the femoral head with viable bone adjacent to other areas infiltrated by GCs [56]. However, it is still unclear whether the severity of bone involvement is related to the extent of the infiltration by GCs or to other pathological mechanisms [57,58]. Radiologically, there is the characteristic appearance of 'worm-eaten' bone tissue [55].

For diagnostic purposes, BMD by dual-energy X-ray absorptiometry (DXA), bone marrow biopsy, axial skeleton X-rays and the search for potential malignant tumors, should be assessed. In adults, it is recommended a DXA of the lumbar spine and left and right hip. Magnetic resonance imaging (MRI) is the radiological gold standard for the skeletal involvement monitoring [59].

Under the age of 19, the radiological evaluation of the tibiae is recommended. However, under the age of 9, it is very difficult to assess the bone marrow infiltration. In addition, children should be examined radiologically every year [60].

The pattern of infiltration provides information on the GD severity. At this regard, there are two typical pictures: type A (homogeneous) and type B (inhomogeneous); the latter is less reversible and with a poor outcome [61].

Another analysis to assess the extent of BM infiltration is a quantitative chemical shift imaging (QCSI) technique [62]. It measures the fat fraction of BM in vertebrae. Values below 0.23 are predictive of bone complications [61].

Moreover, certain indices such as the Dusseldorf score (performed in the MRI examination), especially in adults, together with bone marrow burden (BMB) and with vertebra disc ratio (VDR), could be useful to monitor the disease gravity and the response to pharmacological treatment [63].

Another index, called Italian Gaussi-I severity scoring system, categorizes various BM infiltration scores for stage GD [64].

The scintigraphy with the lipophilic cationic agent 99m Tc-sestamibi [65] is another diagnostic technique to study GD [66,67].

Finally, there is another index for the estimation of GD severity based on hematological, visceral and bone manifestations. This index is called disease severity scoring system (GD1-DS3) [68].

About the clinical management of GD adult patients' goals include the mobility increase, the bone density enhancement and the pain and bone involvement reduction. On the contrary, in pediatric patients, the objectives are the achievement of the ideal peak skeletal mass or the normalization of growth [69].

Skeletal health can be improved with common measures, including an adequate calcium and vitamin D intake by food or oral supplements and the correct management of the pain and of the orthopedic complications [70]. A forward initiation of ERT GD-specific is crucial to optimize outcomes and to prevent irreversible skeletal complications. The latter include the unpredictable onset of AVN. Although there is not an apparent correlation between AVN and GD severity, it has been observed that patients treated 2 years or more after the GD diagnosis, reported more bone crises compared to GD patients timely treated. It is thus hypothesized that bone crises may be an intermediate stage of disease progression and/or an indicator of the future development of AVN [71–73].

Studies in animal models have discovered that a GD1 prolonged nontreatment leads to alterations in the bone microvasculature and may limit the efficacy of ERT for the AVN prevention, due to the chronic inflammation [74]. The risk of AVN after having started ERT in splenectomized patients, is more than two times greater than the non-splenectomized ones. In the past, the splenectomy was commonly performed to control GD1 signs and symptoms, including severe cytopenias and abdominal complaints [75]. To date, it should only be considered in exceptional circumstances [75].

In GD, fractures have a negative impact on quality-of-life [76]. In fact, it is important to timely diagnose the osteopenia because it represents a risk factor for fractures in GD1, as it is highly prevalent in symptomatic patients [77,78]. For the fracture risk assessment, the Gaucher risk assessment for fracture (GRAF) score is used because it seems to be a good predictive index. It is recommended to apply GRAF score at the time of initiation of therapy. The interval between the diagnosis and the start of the treatment is included in GRAF score. The post-treatment fracture risk may help the physician to decide whether to start the treatment or to continue a planned follow-up.

## 4. GD in Dentistry

The GD bone involvement occurs also in the oral cavity and often represents the most disabling aspect of the disease, worsening the quality-of-life [79,80].

Oral and dental manifestations are frequently asymptomatic, in fact, they are incidentally detected through routine dental radiographs or orthopantomography survey [81].

The main features include the GCs infiltration in the jaw, the spontaneous bleeding and the post oral surgery infections [54]. The oral cavity involvement occurs in 75% of GD patients and recent advances in imaging have revealed that the 90% of patients with GD type 1 or 3 have one or more bone manifestations [82]. Therefore, orthopedic maxillofacial prostheses are the only solution to replace bone necrosis and lytic changes [83–85] (Table 3).

- *Radiographic signs in maxilla e mandibular arches*

A loss of the normal trabecularization, probably related to the GCs infiltration, can affect both arches, the maxilla and the mandible [86]. The mandible and maxilla show diffuse osteoporosis or other radiographic signs, typical of thalassemia major and sickle cell anemia. Radiographically, the GD detectable signs are the dislocation of the mandibular canal, the dental or root resorption, the circumscribed radiolucent lesions that appear as cysts and tumors in the mandible with a "soap bubble" appearance (10.8%). The latter are mainly sited in the premolar–molar regions which have a greater amount of medullary bone [87]. Due to these processes, the thinning of the cortical or of the lamina dura and the obliteration of the maxillary sinus can occur (Figure 2).

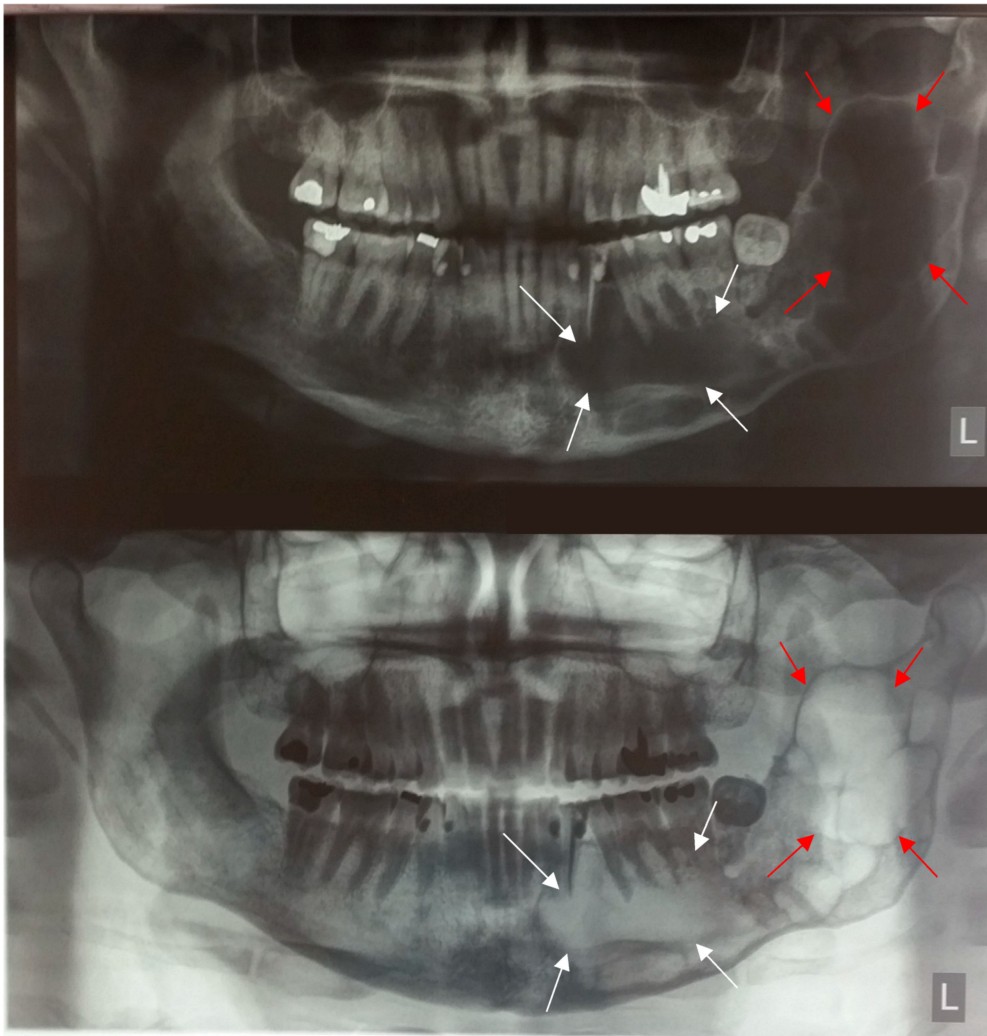

**Figure 2.** Orthopantomografy of a 47-year-old Gaucher disease patient. The white arrows indicate the typical "soap-bubble" appearance in premolar-molar regions. The red arrows indicate the mandibular involvement.

There are several types of bone manifestations in maxillofacial region in association with GD, particularly extra-gnathic bone pathology, which are divided into [86]:

1. Focal disease—characterized by irreversible lesions such as osteonecrosis and osteosclerosis;
2. Local disease—characterized by reversible abnormalities adjacent to the large medullae, such as cortical thinning and bone defects;
3. Generalized osteopenia [88].

Despite the objective radiographic signs of all these processes, the cortical bone remains intact [81,83,89].

The localized rarefaction is the more frequent clinical finding in GD type III, which manifests itself by pseudocystic radiolucent lesions, thinning of the cortex, anodontia and dental abnormalities, whereas, the enlargement of the medullary spaces is a clinical finding in GD type I [83,90]. At the same time, there may be the temporary bone regeneration in post-extraction sites [89], detected by the presence of radiopaque areas in the mandible, which can be noted through imaging techniques. These alterations should be ascribed to osteosclerotic reaction induced by the GCs infiltration [91]. In fact, although root resorption processes are evident, they may be benign and/or they may regress spontaneously. To ascertain the GD diagnosis, some authors have performed biopsies of the jaw to evaluate

the GCs presence. However, if GD is suspected, an enzymatic assay is the most appropriate test to make the diagnosis [92].

- *Other common oral manifestations*

In GD younger patients, a strong correlation has been found between the delayed permanent dentition and the bone involvement, in absence of amyloidosis and/or other bone pathologies [81,93,94]. This eruptive delay is concomitant with a slowed down peak in bone mass and in height. Nevertheless, the growth recovery can occur even without the ERT administration [95].

In all GD patients, other manifestations in the oral cavity, although rare, could be the involvement of the soft tissues of the mouth, the salivary flow reduction [96], or the mono-or bilateral pigmentation on the inner surface of the buccal mucosa, as well as the generalized pigmentation of the face, the lips, the neck and the hands [97–99].

Another sign of GD is the involvement of the lymph nodes, including predominantly the submandibular ones. In particular, it is characterized by their enlargement often due to a secondary process, such as an infection or a transient inflammation [100].

- *Coagulation alterations*

An important clinical sign is the tendency to bleed, which is frequently shown through the presence of petechiae [101]. Its causes include: (i) thrombocytopenia due to hypersplenism, (ii) abnormalities either in the coagulation cascade [102] or (iii) in platelet function [103].

Hollak et al. [102] treated with ERT a group of GD patients, who underwent to a coagulation evaluation and fibrinolysis parameters. The authors showed that GD patients, suffering from platelet dysfunction, should be treated with ERT, before undergoing dental procedures. In fact, after ERT, the GD patients presented a partial correction of coagulation disorders and, in particular, the major reduction in the bleeding was observed after 12 months of ERT. Platelet transfusions, before the dental procedures, are recommended with a high risk of bleeding.

In GD patients, several coagulation factor deficiencies have been described, including factors II, V, IX, X and XI [101,102,104]. ERT seems to increase platelet counts and to improve the coagulative function.

- *Dentistry clinical management*

The dentist must be prepared for the likelihood of an excessive bleeding during dental procedures, such as extractions, curettage and surgery [81,104]. Therefore, it is advisable to perform a prior assessment of a possible coagulation deficit and of an altered platelet aggregation, monitoring prothrombin time (PT), partial thromboplastin time (PTT), the bleeding time and the platelet count [84]. The use of sutures is recommended in these patients.

An important aspect that should not be underestimated is the increased risk of postoperative oral infections in GD patients [105,106], especially if they are not treated with ERT [107] or if they had previously undergone the splenectomy. Therefore, in a GD patient, before the dental extraction, it should be advisable to monitor the coagulative cascade due to the high risk of bleeding. Therefore, to assess the risk of spontaneous bleeding, it is recommended to prescribe: platelet count blood test, international normalized ratio (INR), PT and PTT. If these parameters are predictive of a spontaneous bleeding, the oral surgeon is ready to apply, in the surgery site, the following procedures:

1. Hemostatic compression maneuvers;
2. Sutures;
3. Ice;
4. Tamponade with sterile gauzes;
5. Use local hemostatic drugs.

Moreover, in order to reduce post-operative oral infections, an antibiotic prophylaxis should be recommended from 3 days prior to 3 days after the dental extraction [108].

Before surgery, to prevent the risk of intraoperative complications, it is advisable to carry out an orthopanoramic X-rays or an endoral radiographic control examination.

The post-operative recommendations, on the other hand, include, for the first day, to sleep with two pillows, to avoid efforts that could induce the rupture of the clot or the loss of the sutures, food with seeds, drugs or food that interfere with coagulative cascade, such as grapefruit. Moreover, it may be advisable to avoid hot or hard foods, preferring a cold and liquid diet, and not to brush teeth with toothpaste.

To heal the post-extractive alveolus, GD patients can help themselves with chlorhexidine sprays.

To prevent the early onset of periodontitis in GD patients [101], oral hygiene appropriate instructions and scaling and root planning might be helpful. The co-presence of periodontal disease in GD patients has been described, especially in women [109]. This pathological condition is often improved by bisphosphonates therapy as it seems to reduce the local inflammation and to exert an anti-bone reabsorbing effect [110]. On the other hand, the long-term therapy based on bisphosphonates is associated with a higher risk of the osteonecrosis of the jaw (ONJ); therefore, it is recommended the monitoring of this pharmacological therapy duration [111].

In addition to pharmacological treatment, a soft diet can be recommended in GD patients to avoid the possibility of pathological fractures in the jaw [112,113].

In a survey on Ashkenazi Jewish origin patients, correlations were observed between the gingival index, the decayed missing and obturated surfaces index and the GD clinical signs. This unexpected observation could be due to an increased awareness of the infection [112]. In the presence of GD, an adequate and correct oral hygiene to prevent the odontogenic infections and the secondary osteomyelitis in the jaw should be emphasized.

Finally, in GD patients it is recommended to perform periodic clinical and imaging examinations [114,115].

**Table 3.** GD manifestations and oral disease.

| Type of the Study | Authors | Year | Number of Patients | Description of the Study | Findings |
|---|---|---|---|---|---|
| Case report | Horwitz et al. [101] | 2007 | 1 GD patient | A case report of GD patient with recurrent gingival hemorrhage and toothache | ↑ Resolution of periodontal signs and symptoms by oral hygiene motivation and instructions, scaling, root planning and access flap therapy. |
| Observational | Fischman et al. [104] | 2003 | 350 GD patients | Complete oral and periodontal examination in addition to a routine hematological evaluation. | ↓ Carious lesions in GD patients compared to healthy carriers. |
| Observational | Renvert et al. [109] | 2011 | 778 GD patients | Periodontitis was defined by alveolar bone loss from panoramic radiographs. | Association between osteoporosis and periodontitis only confirmed in women. |
| Case report | Hall et al. [112] | 1985 | 1 GD patient | Affection of the mandible. | ↓ Levels of GCase helps the dentist in the GD diagnosis. |

Abbreviations: AJ, Ashkenazi Jews; AVN, avascular necrosis; BMD, bone mineral density; DS3, disease severity scoring system; ERT, enzyme replacement therapy; Gcase, glucocerebrosidase; GCs, Gaucher cells; GD, Gaucher Disease; MRI, Magnetic Resonance Imaging; MSCs, mesenchymal stromal cells; QCSI, quantitative chemical shift imaging; S-MRI, Spanish- Magnetic Resonance Imaging score; VDR, vertebra disc ratio; ↓, Decreased; ↑, Increased.

## 5. Pharmacological Treatment

There are two different therapeutic approaches to the GD: the ERT and the substrate reduction therapy (SRT) [115].

ERT consists in an intravenous infusion of the enzyme glucocerebrosidase. The Food and Drug Administration (FDA), in 2022, approved three ERTs: imiglucerase, velaglucerase alfa and taliglucerase alfa [116]. Their use was approved for GD type 1 and type 3 without brain involvement, but they are not suitable for GD type 2 and type 3 with brain involvement, because they do not pass through the blood–brain barrier [115]. The purpose of ERT is to slow down the accumulation of the metabolites and prevent further organ damages. ERT acts on the symptoms, but not on the underlying genetic cause. Furthermore, antibodies against the ERT can be developed and, therefore, the treatment could result ineffective [115,116].

Instead, SRT consists in the oral administration of the substrates involved in the glycosphingolipid biosynthesis. The purpose of this therapy is to inhibit the accumulation of toxins and metabolites that can cause organs and tissues damages. The FDA approved two different SRT drugs, namely eliglustat and miglustat. Eliglustat does not pass through the blood–brain barrier, so it is mostly used for the treatment of GD type 1 [117]. The efficacy of eliglustat was considered similar to ERT and greater than miglustat [118]. On the other hand, miglustat can effectively be used for GD type 2 and 3, but, currently, its use is approved only for mild or moderate forms of GD type 1 in adults [115]. The main drawback induced by SRT is the inhibition of other metabolic pathways, such as intestinal disaccharidases, which leads to gastrointestinal disorders [119,120].

Other new therapeutic strategies are represented by chaperone therapy, allogeneic hematopoietic stem cell transplantation (allo-HSCT) and gene therapy [116].

Chaperone therapy restores GCase activity and corrects the misfolding of enzymes. Ambroxol, an inhibitor of GCase, is considered one of the potential candidates to the chaperone therapy [121]; it seems to have positive effects on adult neurological manifestations [122] and on pediatric skeletal and hematological alterations [123]. Unfortunately, it is a specific mutation-therapy [116].

Allo-HSCT is a therapeutic strategy in which hematopoietic stem cells are transferred from a healthy donor to an affected recipient, in order to restore the physiological enzyme activity [116]. Allo-HSCT seems to operate either the improvement of visceral and skeletal alterations and the stabilization of neurological manifestations [124–127].

Another therapeutic approach is represented by gene therapy (GT), in which the gene is modified in order to treat the disease. It is distinguished into in vivo and ex vivo. In the first technique, the vector is injected into the patient; however, in the second one the vector is injected into the patient's cells and then they are retransplanted [116]. Over the years in the ex vivo technique, the gammaretroviral and the lentiviral vectors were tested [128,129]. The most promising clinical trial is represented by GuardOne, based on a lentiviral vector [130,131]. Currently, there is no clinical trial in which is tested in vivo GT for the GD treatment [116].

## 6. Conclusions

Even if GD is a rare pathological condition, the physician should not underestimate the typical clinical signs and the systemic manifestations which lead to its diagnosis. In fact, an early diagnosis allows GD patients to start the therapy on time and to reduce its clinical manifestations. The main GD systemic clinical manifestations involve the hematopoietic apparatus, the nervous system, the liver, the cardiovascular system, the kidney and the bone tissue. Although the incidence of oral involvement is one of the less common manifestations of GD, the presence of lesions especially in the jaw, is a constant feature in orthopantomografy or cone beam computed tomography. The dentist should intercept these lesions to make a diagnosis of the GD and to counteract the possible oral and dental complications. The hematological or coagulation abnormalities associated with GD may complicate the oral diseases management. An adequate evaluation of coagulative and

fibrinolysis parameters should be considered prior to dental surgery procedures. Moreover, it is useful to schedule a follow-up session and to start a careful monitoring of the possible complications in the post-operative period.

**Author Contributions:** Conceptualization, A.N.; methodology, E.S. and V.P.; writing—original draft preparation, M.B., G.M., M.D.L., E.S., V.P. and G.D.; writing—review and editing, P.B. and A.N.; visualization, V.P. and R.H.; supervision, P.B. and A.N. All authors have read and agreed to the published version of the manuscript.

**Funding:** This research received no external funding.

**Institutional Review Board Statement:** Not applicable.

**Informed Consent Statement:** Not applicable.

**Data Availability Statement:** Not applicable.

**Acknowledgments:** We would like to thank Gabriella Venafro for English language revision and we are in debt to Enzo Ventucci for his clinical assistance.

**Conflicts of Interest:** The authors declare no conflict of interest.

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
