# Peer review of "Gaucher Disease in Internal Medicine and Dentistry"

_applsci, doi:10.3390/app13064062_

Round 1

Reviewer 1 Report

Some original radiographs of the affected patients are necessary.  Also provide a clinical picture.  Both of these suggestions would assist clinicians to diagnose this rare condition.  "the circumscribed radiolucent lesions that appear as  cysts and tumors in the mandible with a “soap bubble” appearance"  - sentences such as this would be more intelligible with a clear radiographic example.  Also summarise the main points- what should a dentist do if such a patient requires an extraction - blood coagulation tests?  pre-operative radiographs?  - the practical advice is a bit lost in the text.  I highlighted a few minor errors in the English.

Author Response

Prof. Annalisa Noce

Dr. Manuela Di Lauro

Rome,17 March 2023

Dear Editor,

all the corrections have been written in red colour in the revised version of the manuscript. We reviewed the entire manuscript according to the reviewer’s comments.

We would like to thank reviewer #1 for his/her comments.

He/she wrote:

  • Some original radiographs of the affected patients are necessary.  Also provide a clinical picture.  Both of these suggestions would assist clinicians to diagnose this rare condition.  "the circumscribed radiolucent lesions that appear as cysts and tumors in the mandible with a “soap bubble” appearance" - sentences such as this would be more intelligible with a clear radiographic example

We added a radiographic image (Figure 2), in which we described the soap bubble appearance indicated by arrows.

  • Also summarise the main points- what should a dentist do if such a patient requires an extraction - blood coagulation tests?  pre-operative radiographs?  - the practical advice is a bit lost in the text. 

We added practical advices in the section 4, subheading “Dentistry clinical management”.

  • I highlighted a few minor errors in the English.

We corrected the minor errors.

Best regards,

Annalisa Noce

Manuela Di Lauro

Reviewer 2 Report

The authors present an exhaustive approach of Gaucher Disease, the most common lysosomal storage disease, focusing on the field of internal medicine and dentistry. Although the documentation of this work has been very extensive, there are several aspects that need to be corrected.

Section 1. Authors should provide an explanation of why the review is focused on internal medicine and dentistry. The approach is fragmented. I just to give examples: that the oral manifestations of this disease are very frank, that the anatomic site with more diversity of lesions, that the general practitioner of dentistry can consult with the specialist in internal medicine to face this patients, etc.

Section 1, lines 69-71. The authors mention that the objective is to "analyze" the clinical manifestations, but after reading the manuscript there is no analytical approach, only a recapitulation and description of the characteristics. Therefore, I request that the verb "analyze" be changed to another verb that truly reflects the product of this work.

Section 2. Material and Methods. Although the title does not specify that this is a systematic review, in lines 69-70 they state that their objective is a review, therefore it is prudent to clarify in this section what their criteria were for incorporating the different articles they selected for this work. This also has an impact on the design of their tables, especially in the first column "type of study", since absolutely all the articles are "Human study"! That is, there is no "Experimental/in vivo model", "Experimental/in vitro model". This is important, because the level of scientific evidence is different for each design. Therefore, and at least to strengthen the findings summarized in the tables, the authors should incorporate in the column the type of study and not only "Human study", i.e.: cases-controls, cross-sectional, case report, cases series, cohort, clinical trial, etc.

None of the three tables are referenced in the text, they are supposed to be an auxiliary for the condensation of the information. Please reference them in the text.

Authors mentioned in the tables should be assigned the corresponding reference number within the tables. 

Section 3.1, second paragraph, lines 158-160. Refer to De Brito et al., according to your listing it is reference number 14, and therefore correct the reference number you assigned to Smith et al. which seems to correspond to reference number 16; alternatively, cite in the appropriate place.

Section 3.1, Paragraph 4, lines 164-165. This information is outdated (year 1977), the same authors mention that the affected macrophages may be an M2 subpopulation (line 51-52), the term reticuloendothelial is obsolete, it is preferable to use mononuclear phagocytic system; I suggest deleting this section.

Section 3.1, first paragraph, lines 151-153, it is doubtful that only 3 cases since 1973 (Ref 14), 1978 (Ref 16) and 1969 (Ref 17) have had renal involvement. In fact in reference 33 (year 1998) it mentions another case. Please restructure your sentence.

Section 4. The findings described in this section are not supported by Table 3. There are few studies that refer to stomatological findings. The authors should show in this table only findings similar to those of Horwitz (Ref 101), Fischman (Ref 105), Renvert (Ref 109) or Hall (Ref 112).

General remarks

- The approach to the subject is fractioned and is not integrative, the authors have not linked historical information with the current state of information.

- The authors do not explain the method of selection of their literature and combine different study designs in their tables. As a result, in their findings, they mention very peculiar features of case reports and these are at the same level as epidemiological studies.

- They do not explain a link between the two disciplines (internal medicine and dentistry) that reasonably justifies the work; of course this link exists.

- The tables have no order, and above all they do not highlight the clinical findings of each of the sections the authors address, this is most striking in the dentistry section in which only 4 authors cover the focus of the table.

- The abstract is constructed by different fragments of the text, they are almost equal sections.

Minor comments

In the section "Pharmacological treatment", line 435, the number "5." is missing, in the same way as in the rest of the sections.

Section 3. Line 108. Correct the word "omozygus", it should be "homozygous".

Section 3, line 144, remove the middle hyphen between "apolipo-protein E".

Reference 15. The name of the journal is not abbreviated, it should be: "J Pathol" (line 515), also the doi is underlined, as in reference 67 (line 640), correct the format or remove the hyperlink.

Section 5. line 449, Correct the word “Eligustat”, it should be “Eliglustat”.

Correctly abbreviate the journals in references 67 and 88.

Author Response

Prof. Annalisa Noce

Dr. Manuela Di Lauro

Rome,17 March 2023

Dear Editor,

all the corrections have been written in red colour in the revised version of the manuscript. We reviewed the entire manuscript according to the reviewer’s comments.

We would like to thank reviewer #2 for his/her comments.

He/she wrote:

  • Section 1 Authors should provide an explanation of why the review is focused on internal medicine and dentistry. The approach is fragmented. I just to give examples: that the oral manifestations of this disease are very frank, that the anatomic site with more diversity of lesions, that the general practitioner of dentistry can consult with the specialist in internal medicine to face this patients, etc.

We added information regarding the link between internal medicine and dentistry in GD.

  • Section 1 lines 69-71. The authors mention that the objective is to "analyze" the clinical manifestations, but after reading the manuscript there is no analytical approach, only a recapitulation and description of the characteristics. Therefore, I request that the verb "analyze" be changed to another verb that truly reflects the product of this work.

We changed the verb to “discuss”.

  • Section 2 Material and Methods. Although the title does not specify that this is a systematic review, in lines 69-70 they state that their objective is a review, therefore it is prudent to clarify in this section what their criteria were for incorporating the different articles they selected for this work. This also has an impact on the design of their tables, especially in the first column "type of study", since absolutely all the articles are "Human study"! That is, there is no "Experimental/in vivo model", "Experimental/in vitro model". This is important, because the level of scientific evidence is different for each design. Therefore, and at least to strengthen the findings summarized in the tables, the authors should incorporate in the column the type of study and not only "Human study", i.e.: cases-controls, cross-sectional, case report, cases series, cohort, clinical trial, etc.

We added a new column to describe the type of the studies.

  • None of the three tables are referenced in the text, they are supposed to be an auxiliary for the condensation of the information. Please reference them in the text.

Authors mentioned in the tables should be assigned the corresponding reference number within the tables. 

We added the reference numbers in the tables.

  • Section 3.1, second paragraph, lines 158-160. Refer to De Brito et al., according to your listing it is reference number 14, and therefore correct the reference number you assigned to Smith et al. which seems to correspond to reference number 16; alternatively, cite in the appropriate place.

We corrected the references in the text.

  • Section 3.1, Paragraph 4, lines 164-165. This information is outdated (year 1977), the same authors mention that the affected macrophages may be an M2 subpopulation (line 51-52), the term reticuloendothelial is obsolete, it is preferable to use mononuclear phagocytic system; I suggest deleting this section.

We deleted the sentence, as suggested.

  • Section 3.1, first paragraph, lines 151-153, it is doubtful that only 3 cases since 1973 (Ref 14), 1978 (Ref 16) and 1969 (Ref 17) have had renal involvement. In fact in reference 33 (year 1998) it mentions another case. Please restructure your sentence.

We rephrased the sentence.

  • Section 4. The findings described in this section are not supported by Table 3. There are few studies that refer to stomatological findings. The authors should show in this table only findings similar to those of Horwitz (Ref 101), Fischman (Ref 105), Renvert (Ref 109) or Hall (Ref 112).

We showed in the table the studies with oral findings according to the reviewer’s comment.

  • The authors do not explain the method of selection of their literature and combine different study designs in their tables. As a result, in their findings, they mention very peculiar features of case reports and these are at the same level as epidemiological studies

We added in the materials and methods section the study designs discussed.

  • They do not explain a link between the two disciplines (internal medicine and dentistry) that reasonably justifies the work; of course this link exists.

We added information regarding the link between internal medicine and dentistry in GD.

  • The tables have no order, and above all they do not highlight the clinical findings of each of the sections the authors address, this is most striking in the dentistry section in which only 4 authors cover the focus of the table.

We showed in the table the studies with oral findings according to the reviewer’s comment.

  • The abstract is constructed by different fragments of the text, they are almost equal sections.

We revised the abstract according your comments.

  • In the section "Pharmacological treatment", line 435, the number "5." is missing, in the same way as in the rest of the sections.

We inserted the number of paragraph.

  • Section 3. Line 108. Correct the word "omozygus", it should be "homozygous".

We corrected the typo.

  • Section 3, line 144, remove the middle hyphen between "apolipo-protein E".

We corrected the typo.

  • Reference 15. The name of the journal is not abbreviated, it should be: "J Pathol" (line 515), also the doi is underlined, as in reference 67 (line 640), correct the format or remove the hyperlink. 

We renamed the journal name as “J Pathol” and we deleted the hyperlink of the two references.

  • Section 5. line 449, Correct the word “Eligustat”, it should be “Eliglustat”.

We corrected the typo.

  • Correctly abbreviate the journals in references 67 and 88.

We inserted the journal abbreviations.

Best regards,

Annalisa Noce

Manuela Di Lauro

Reviewer 3 Report

Report of manuscript: Gaucher Disease in Internal Medicine and Dentistry

Author performed a narrative review of GD in three aspects including: Internal medicine, dentistry and pharmacological management.

Overall, many minor paragraphs are including in each topic. Author should divided in to sub-heading. 

Page 2: The first figure should be in high quality. Number and description of figure is missing.

Table 1: What is the arrow represent? Arrow represents increasing or decreasing of any values must be explained in the abbreviation under the table as well.

Page 12:Example picture of the Oral manifestation and  Radiographic appearance representing pathological changes in Maxilla and mandible must be added in this manuscript for better understanding.

Page 12: GD in dentistry. Author should rewrite the paragraph into well separated topic in dentistry branches like dental surgery, oral manifestation, radiographic appearance.

Line 399: Please see the format.

Table 3: The heading of this table should begin with the new page.

Page 18: Pharmacological management is too short. If author would like to include this topic, author should add more information and new finding in this topic. Only two references are added to this topic. Please add more detail. 

Author Response

Prof. Annalisa Noce

Dr. Manuela Di Lauro

Rome,17 March 2023

Dear Editor,

all the corrections have been written in red colour in the revised version of the manuscript. We reviewed the entire manuscript according to the reviewer’s comments.

We would like to thank reviewer #3 for his/her comments.

  • Overall, many minor paragraphs are including in each topic. Author should divided in to sub-heading. 

We added the subheading in the paragraphs.

  • Page 2: The first figure should be in high quality. Number and description of figure is missing.

We added the number and description of figure.

  • Table 1: What is the arrow represent? Arrow represents increasing or decreasing of any values must be explained in the abbreviation under the table as well.

We added the explanation of the arrows

  • Page 12: Example picture of the Oral manifestation and Radiographic appearance representing pathological changes in Maxilla and mandible must be added in this manuscript for better understanding.

We added the image (Figure 2).

  • Page 12: GD in dentistry. Author should rewrite the paragraph into well separated topic in dentistry branches like dental surgery, oral manifestation, radiographic appearance.

We added the subheading in the paragraphs

  • Line 399: Please see the format.

We checked the format.

  • Table 3: The heading of this table should begin with the new page.

The heading of the table starts at the new page.

  • Page 18: Pharmacological management is too short. If author would like to include this topic, author should add more information and new finding in this topic. Only two references are added to this topic. Please add more detail. 

We added more studies in the pharmacological treatment.

Best regards,

Annalisa Noce

Manuela Di Lauro

Round 2

Reviewer 2 Report

The authors have made the suggested corrections in the tables, they have organized the different sections which makes the overall text more reader-friendly, they have incorporated new images, they have also modified or restructured ideas in several sections of the text and justified the link between the two disciplines for the approach to this disease, in addition, they have added more references to their review. I consider these to be sufficient changes to continue with their publication process.